# Noisy-Channel Minimum Bayes Risk Decoding

**Yusuke Sakai** [1]  **Hidetaka Kamigaito** [1]  **Taro Watanabe** [1]

## Abstract

Minimum Bayes Risk (MBR) decoding yields more robust and higher-quality text generation than maximum a posteriori (MAP) decoding by selecting hypotheses that maximize expected utility over sampled pseudo-references. However, there exists a discrepancy in the design: hypothesis selection calculates expected utility scores conditioned on given pseudo-references, while commonly used evaluation metrics, e.g., BLEU and COMET, are asymmetric. Therefore, it is important to consider both hypothesis-to-reference and reference-to-hypothesis directional effects. In this study, we introduce a noisy channel decomposition of MBR decoding that naturally incorporates bidirectional effects to account for these asymmetries. We decompose MBR decoding into four interacting components: hypothesis-to-reference likelihood, reference-to-hypothesis likelihood, hypothesis prior, and reference prior. This decomposition provides a unified interpretation of existing MBR variants and enables metric- and task-specific interpretability by isolating the contribution of each channel. Our comprehensive analysis reveals that channel-wise contributions exhibit distinct characteristics across metrics while remaining consistent across tasks, and suggests that appropriate channel weighting may lead to improvements over original MBR decoding.

## 1. Introduction

Decoding in text generation systems typically relies on heuristic strategies such as greedy decoding, beam search, and sampling methods, which approximate maximum a posteriori (MAP) decoding by selecting the most probable hypothesis under the model's output distribution. However, MAP decoding often fails to align with downstream evaluation metrics and human preferences (Koehn & Knowles, 2017; Eikema & Aziz, 2020). Minimum Bayes Risk (MBR) decoding (Goel & Byrne, 2000; Kumar & Byrne, 2004; Eikema & Aziz, 2020) yields more robust and higher-quality text generation than MAP decoding by directly optimizing expected utility with respect to an evaluation metric, such as BLEU (Papineni et al., 2002) or BERTScore (Zhang et al., 2020). Rather than selecting the most likely hypothesis, MBR decoding selects **the hypothesis that maximizes expected utility over a set of sampled pseudo-references**.

However, the hypothesis selection formulation of original MBR decoding exhibits a fundamental mismatch with commonly used evaluation metrics, such as BLEU. These metrics are inherently asymmetric and do not treat hypotheses and references interchangeably. Their scores also depend on the direction of comparison. Consequently, effective hypothesis selection should consider not only reference-to-hypothesis relationships, but also **hypothesis-to-reference interactions**, as well as priors over hypotheses and references. Such directional asymmetries and prior effects are not explicitly captured in existing MBR formulations, highlighting a fundamental limitation of the MBR decoding.

In this study, we introduce a noisy-channel-based decomposition of MBR decoding that naturally accounts for such bidirectional interactions. Our formulation decomposes MBR decoding into four interacting components: the hypothesis-to-reference likelihood, the reference-to-hypothesis likelihood, the hypothesis prior, and the reference prior. This decomposition provides a probabilistic interpretation that explicitly models bidirectional effects between hypotheses and references, thereby flexibly aligning the expected utility. Furthermore, this decomposition enables fine-grained interpretability. By isolating the contribution of each channel, our framework allows us to analyze which components are most influential for a given metric or task. This perspective provides a unified interpretation of existing MBR variants (Jinnai et al., 2024) and clarifies how MBR decoding implicitly emphasizes different probabilistic factors.

In our empirical analysis, we demonstrate that channel-wise weighting based on our decomposition effectively captures the role of each component in MBR decoding. Our results show that decoding performance is primarily influenced by the choice of evaluation metric as the utility function and

[1]Nara Institute of Science and Technology, Nara, Japan. Correspondence to: Yusuke Sakai <sakai.yusuke.sr9@is.naist.jp>.

*Proceedings of the 43rd International Conference on Machine Learning*, Seoul, South Korea. PMLR 306, 2026. Copyright 2026 by the author(s).

is task-agnostic. In particular, when combined with appropriate channel weighting, stable metrics, e.g., BERTScore, consistently yield performance gains. These results highlight the practical benefits of our formulation, showing that a theoretically grounded decomposition can simultaneously improve interpretability and decoding performance.

## 2. Background and Related Work

**Maximum a Posteriori Decoding**   Decoding for text generation systems is to find the optimal output text $h$ for a given input text $x$ based on a decision strategy. Since the task is intractable in that $h$ is searched from the infinite output space $\mathcal{Y}$, we usually find the optimal solution using the limited search space $\mathcal{H} \subseteq \mathcal{Y}$ instead.

One of the most popular decision strategies is maximum a posteriori (MAP) decoding, such as that used in greedy and beam search. MAP decoding searches for an optimal output $h^{\mathrm{MAP}} \in \mathcal{H}$ according to its output probability $P(h|x; \phi)$ calculated by a text generation model $\phi$ as follows:

$$h^{\mathrm{MAP}} = \arg\max_{h \in \mathcal{H}} P(h|x; \phi). \qquad (1)$$

Since MAP decoding solely relies on the generation probability by the model $\phi$, the high-probability output $h^{\mathrm{MAP}}$ is not guaranteed to be the best in terms of quality for a target task. Specifically, even if we obtain a sequence with a higher probability by increasing the beam size, such sequences may not be optimal for task performance, leading to sequences with errors, e.g., empty sequences, $n$-gram repetitions, and copies of the input sequence (Koehn & Knowles, 2017; Ott et al., 2018; Eikema & Aziz, 2020).

**Minimum Bayes Risk Decoding**   Minimum Bayes risk (MBR) decoding is designed to alleviate the problems of MAP decoding (Kumar & Byrne, 2004; Eikema & Aziz, 2020) to find high-quality outputs by considering task-specific scores and is formulated as follows:

$$h^{\mathrm{MBR_{true}}} = \arg\max_{h_i \in \mathcal{H}} \mathbb{E}_{r \sim P(r|x)} \left[ f_\theta(h_i, r) \right], \qquad (2)$$

where $P(r|x)$ is the true probability of $r$ given $x$. $f_\theta : \mathcal{Y} \times \mathcal{Y} \to \mathbb{R}$ is a utility function parameterized by $\theta$ that measures the preference of generation $h$ given a reference $r$, which is defined as $h \succeq h' \iff f_\theta(h, r) \geq f_\theta(h', r)$ where $\succeq$ denotes the preference relation[1]. The true probability $P(r|x)$ is unknown and then replaced with the generation

model probability $P(r|x; \phi)$ as follows:

$$h^{\mathrm{MBR}_\phi} = \arg\max_{h_i \in \mathcal{H}} \mathbb{E}_{r \sim P(r|x;\phi)} \left[ f_\theta(h_i, r) \right]. \qquad (3)$$

Since enumerating all possible outputs in $\mathcal{Y}$ is infeasible, the expectation of the utility function is estimated by the Monte Carlo method (Eikema & Aziz, 2022) on pseudo-references $\mathcal{R} \coloneqq \{ r_i \in \mathcal{Y} \mid r_i \sim P(r_i|x; \phi) \}$[2], sampled according to the output probabilities of the generation model as follows:

$$h^{\mathrm{MBR_{MC}}} = \arg\max_{h_i \in \mathcal{H}} \frac{1}{|\mathcal{R}|} \sum_{r_j \in \mathcal{R}} f_\theta(h_i, r_j). \qquad (4)$$

**Prior Advances**   Recent studies have primarily focused on improving the efficiency of MBR decoding (Cheng & Vlachos, 2023; Jinnai & Ariu, 2024; Deguchi et al., 2024b; Vamvas & Sennrich, 2024; Trabelsi et al., 2024; Natsumi et al., 2025) or its decoding quality (Jinnai et al., 2024; Daheim et al., 2025; Deguchi & Nagata, 2025). Some studies further apply MBR decoding to specific downstream tasks (Bertsch et al., 2023; Jinnai et al., 2025; Lyu et al., 2025a;b; Deguchi et al., 2023; Kudo et al., 2024; Hayakawa et al., 2025), explore the design of utility functions for various downstream applications (Jinnai, 2025; Eikema et al., 2025; Amrhein & Sennrich, 2022; Naskar et al., 2023; Yan et al., 2024; Vashurin et al., 2025; Soetedjo et al., 2026), or investigate other aspects of MBR decoding (Yang et al., 2024; Amrhein & Sennrich, 2022; Finkelstein et al., 2024). In addition, both theoretical (Kamigaito et al., 2025; Ichihara et al., 2025) and empirical analyses (Müller & Sennrich, 2021; Freitag et al., 2023; Ohashi et al., 2024) have investigated the conditions under which MBR decoding is effective. Moreover, MBR decoding is concerned with selecting the best candidate from a given set of samples. Consequently, while some studies have investigated the choice of sampling methods, effects related to the underlying model architecture are generally treated as outside the scope of MBR decoding itself. Despite these advances, these approaches typically rely on a single reference-to-hypothesis direction. As a result, the potential role of bidirectional dependencies remains implicit and is not explicitly analyzed or effectively utilized.

## 3. Noisy-channel-based Decomposition

### 3.1. Derivation

We derive a joint probability-based reformulation of Equation 4 as follows[3]:

---

[1]For example, BLEU (Papineni et al., 2002) and COMET (Rei et al., 2020; 2022) are often used as utility functions $f_\theta$ in machine translation tasks, while BERTScore (Zhang et al., 2020) is commonly used in summarization and captioning tasks. Such automatic metrics are typically evaluated based on their correlation with human preferences through meta-evaluation studies (Freitag et al., 2024). In this sense, they already provide a reasonably reliable proxy for human judgment.

[2]In a typical application, the hypotheses themselves are treated as the pseudo-references, i.e., $\mathcal{H} = \mathcal{R}$.

[3]We can treat utility functions as non-negative in this normalization because Equation 4 involves an argmax operation and is invariant to additive shifts.

*Table 1.* Summary of the unified interpretation from the viewpoint of our derived $P(r_j|h_i)^\alpha P(h_i)^\beta P(h_i|r_j)^\gamma P(r_j)^\delta$ in Equation 13. $\mathcal{U}(0,1)$ is a uniform distribution ranging from 0 to 1.

| Corresponding Method | | MAP | Original | Ours (NEW) | | | |
|---|---|---|---|---|---|---|---|
| | | | | Conditional | Swap | Inverse | Others |
| Formulation | | $\mathcal{U}(0,1)$ | $P(h_i|r_j)P(r_j)$ | $P(h_i|r_j)$ | $P(r_j|h_i)P(h_i)$ | $P(r_j|h_i)$ | $P(r_j|h_i)^\alpha P(h_i)^\beta P(h_i|r_j)^\gamma P(r_j)^\delta$ |
| Hyperparameters | $\alpha$ | 0 | 0 | 0 | 1 | 1 | |
| | $\beta$ | 0 | 0 | 0 | 1 | 0 | Others |
| | $\gamma$ | 0 | 1 | 1 | 0 | 0 | |
| | $\delta$ | 0 | 1 | 0 | 0 | 0 | |

$$(4) = \arg\max_{h_i \in \mathcal{H}} \sum_{r_j \in \mathcal{R}} f_\theta(h_i, r_j) \quad (5)$$

$$= \arg\max_{h_i \in \mathcal{H}} \sum_{r_j \in \mathcal{R}} \frac{f_\theta(h_i, r_j)}{\Sigma_{y_{j'} \in \mathcal{R}} \Sigma_{h_{i'} \in \mathcal{H}} f_\theta(h_{i'}, y_{j'})} \quad (6)$$

$$= \arg\max_{h_i \in \mathcal{H}} \sum_{r_j \in \mathcal{R}} \frac{f_\theta(h_i, r_j)}{\Sigma_{h_{i'} \in \mathcal{H}} f_\theta(h_{i'}, r_j)}$$

$$\cdot \frac{\Sigma_{h_{i'} \in \mathcal{H}} f_\theta(h_{i'}, r_j)}{\Sigma_{y_{j'} \in \mathcal{R}} \Sigma_{h_{i'} \in \mathcal{H}} f_\theta(h_{i'}, y_{j'})} \quad (7)$$

$$= \arg\max_{h_i \in \mathcal{H}} \sum_{r_j \in \mathcal{R}} P(h_i|r_j)P(r_j). \quad (8)$$

Using a noisy-channel-based decomposition, $P(h_i|r_j) = \frac{P(r_j|h_i)P(h_i)}{P(r_j)}$, we can decompose Equation 8 as follows:

$$(8) = \arg\max_{h_i \in \mathcal{H}} \sum_{r_j \in \mathcal{R}} \sqrt{P(h_i|r_j)^2 P(r_j)} \quad (9)$$

$$= \arg\max_{h_i \in \mathcal{H}} \sum_{r_j \in \mathcal{R}} \sqrt{\frac{P(r_j|h_i)P(h_i)}{P(r_j)} P(h_i|r_j)P(r_j)^2}$$

$$\quad (10)$$

$$= \arg\max_{h_i \in \mathcal{H}} \sum_{r_j \in \mathcal{R}} \sqrt{P(r_j|h_i)P(h_i)P(h_i|r_j)P(r_j)} \quad (11)$$

As shown in Equation 11, we can represent the scoring of MBR decoding as the four distinct probabilistic terms: *hypothesis-to-reference likelihood* $P(r_j|h_i)$, *reference-to-hypothesis likelihood* $P(h_i|r_j)$, *hypothesis prior* $P(h_i)$, and *reference prior* $P(r_j)$. In this paper, to incorporate the inverse direction of metric functions, we calculate $P(r_j|h_i)$ by swapping $r_j$ and $h_i$, i.e., $f_\theta(r_j, h_i)$, as follows:

$$P(r_j|h_i) = \frac{f_\theta(r_j, h_i)}{\Sigma_{r_{j'} \in \mathcal{R}} f_\theta(r_{j'}, h_i)} \quad (12)$$

Here, inspired by the concept of the source channel model (Och & Ney, 2002; Koehn et al., 2003; Li et al., 2004) improving decoding performance by reweighting each prob-

abilistic term, we expand Equation 11 as follows:

$$\arg\max_{h_i \in \mathcal{H}} \sum_{r_j \in \mathcal{R}} \sqrt{P(r_j|h_i)^\alpha P(h_i)^\beta P(h_i|r_j)^\gamma P(r_j)^\delta}, \quad (13)$$

where $\alpha$, $\beta$, $\gamma$, and $\delta$ are hyperparameters for adjusting the importance of each term. This expansion is regarded as a decision-theoretic concept rather than a true probabilistic decomposition and is widely used in source-channel models (Och & Ney, 2002; Yu et al., 2017).

Through this paper, we investigate the characteristics and importance of each decomposed probabilistic term in MBR decoding by changing the hyperparameters. The hyperparameters larger than 1 mean that the corresponding probabilities are strengthened, while those lower than 1 mean that the corresponding probabilities are weakened when deciding the rank of hypotheses. Note that our proposed decomposition mainly involves simple operations such as weighted sums and scaling, and can be computed efficiently in a sequential manner. As a result, it introduces negligible overhead compared to standard MBR.

### 3.2. Unified Interpretation

Based on the formulation in Equation 13, we can interpret various MBR decoding approaches by varying four hyperparameters. Table 1 summarizes the relationship between these approaches under the corresponding hyperparameters. Since the outer square root in Equation 13 does not change the rank of the hypotheses, we ignore this part and discuss the approaches in Table 1 in the following paragraphs:

**MAP** When all hyperparameters are zero, the utility function does not affect the rank of the hypotheses in MBR decoding, and this setting is identical to MAP decoding.

**Original** Since $P(h_i|r_j)P(r_j) = P(h_i, r_j) \propto f_\theta(h_i, r_j)$, this setting corresponds to the original MBR decoding in Equation 4. The prior part $P(r_j)$ is also represented as a generation probability by a model, implicitly like sampling in Equation 4, and explicitly like in the model-based MBR

decoding (Jinnai et al., 2024) as follows:

$$\underset{h_i \in \mathcal{H}}{\arg\max} \sum_{r_j \in \mathcal{R}} P_{\mathrm{MB}}(r_j|x; \mathcal{R}, \phi) f_\theta(h_i, r_j), \quad (14)$$

where $P_{\mathrm{MB}}$ is the normalized output probability over a set of pseudo-references, as follows:

$$P_{\mathrm{MB}}(r|x; \mathcal{R}, \phi) := \frac{P(r|x; \phi)}{\sum_{r_j \in \mathcal{R}} P(r_j|x; \phi)}. \quad (15)$$

**Conditional**   Different from the original MBR decoding, this setting only considers the posterior probability part $P(h_i|r_j)$. The main difference from the commonly used scoring methods in MBR decoding is the existence of normalization. Kamigaito et al. (2025) interpret the MBR decoding with the normalization in a utility function as the Bayes Optimal Classifier (BOC) (Mitchell, 1997) as shown in the following reformulation:

$$\underset{h_i \in \mathcal{H}}{\arg\max} \frac{1}{|\mathcal{R}|} \sum_{r_j \in \mathcal{R}} P(h_i|r_j), \quad r_j \sim P(r_j|x; \phi) \quad (16)$$

$$\approx \underset{h_i \in \mathcal{H}}{\arg\max} \sum_{r_j \in \mathcal{R}} P(h_i|r_j) P(r_j|x; \phi). \quad (17)$$

Therefore, we can understand that our proposed decomposition also covers the traditional ensemble method, BOC as well as other approaches.

**Swap**   This setting is essentially the same as the original MBR decoding. However, its behavior depends on the definition of the posterior probability part $P(r_j|h_i)$. As we explained in the previous section, $P(r_j|h_i)$ in our paper utilizes the inverse direction of metric functions, and thus, it can consider the information not captured in the *Original* and *Conditional* settings. Because some metrics like chrF and BERTScore return the same score by swapping their input, this setting is identical to *Conditional* in such metrics.

**Inverse**   Similar to the relationship between *Original* and *Swap*, this setting is the inverse version of *Conditional*. Since the calculation direction between hypotheses and pseudo-references does not change the probabilistic decomposition, this approach is also classified as a kind of BOC, a method in ensemble learning following Equation 17.

**Others**   This setting is not grounded in any conventional methods. Thinking about the fact that only a few methods are covered by the previous paragraphs against this large parameter space, exploring various parameters in this setting is quite important to find a sweet spot for MBR decoding.

In addition to interpreting and uncovering MBR decoding methods, our formulation in Equation 13 can reveal the strengths and weaknesses of metrics when used as utility functions. We investigate these aspects in our analysis.

## 4. Analysis

We investigate the characteristics and importance of each decomposed probabilistic term in MBR decoding by analyzing the corresponding parameters. Specifically, we address the following two research questions through the experiments:

**RQ1: Do different "utility functions" exhibit consistent or characteristic trends across channels?**

**RQ2: Do different "types of tasks" exhibit consistent channel-wise trends under the same utility function?**

These RQs allow us to answer (i) how the choice of utility function influences MBR decoding performance, and (ii) to what extent channel-wise importance in MBR decoding is task-agnostic, thereby enabling a comprehensive evaluation of the effectiveness and efficiency of the MBR decoding.

### 4.1. RQ1: Channel Importance of Utility Functions

#### 4.1.1. SETTINGS

We conducted experiments on machine translation tasks covering German–English (En↔De), Japanese–English (Ja↔En), and Chinese–English (Zh↔En) language pairs under the WMT'22 (Kocmi et al., 2022) and WMT'23 (Kocmi et al., 2023) general translation tasks, resulting in 12 evaluation scenarios across language directions and datasets. Following Deguchi et al. (2024a), we sampled 256 translation candidates via $\varepsilon$-sampling with $\varepsilon = 0.02$ (Freitag et al., 2023) from M2M100 (FACEBOOK/M2M100_418M) (Fan et al., 2020) and used the same set of candidates as pseudo-references. As utility functions, we compare four evaluation metrics: BLEU (Papineni et al., 2002), chrF (Popović, 2015), COMET (Rei et al., 2020), and BERTScore (Zhang et al., 2020), and report results using the same metrics. We use MBRS (Deguchi et al., 2024a) as the backbone of our experiments, with all other settings following the default configuration, including hyperparameters and language-specific preprocessing. We conduct a grid search over $[0.00, 0.25, 0.50, 0.75, 1.00, 1.25, 1.50, 1.75, 2.00]$ for each channel weight $\alpha$, $\beta$, $\gamma$, and $\delta$. We use a single NVIDIA RTX 3090 GPU for all experiments.

#### 4.1.2. RESULTS

To analyze the importance of individual terms, we conduct a controlled analysis in which one term weight (i.e., $\alpha$, $\beta$, $\gamma$, or $\delta$) is varied while the remaining weights are averaged out. This enables us to quantify the relative contribution of each term under a given utility function. Figure 1 shows the results. Since absolute performance varies substantially across metrics and tasks, as well as in overall score ranges, we focus on capturing relative trends rather than absolute scores. To enable a unified comparison, we report z-scores normalized with respect to the setting where the correspond-

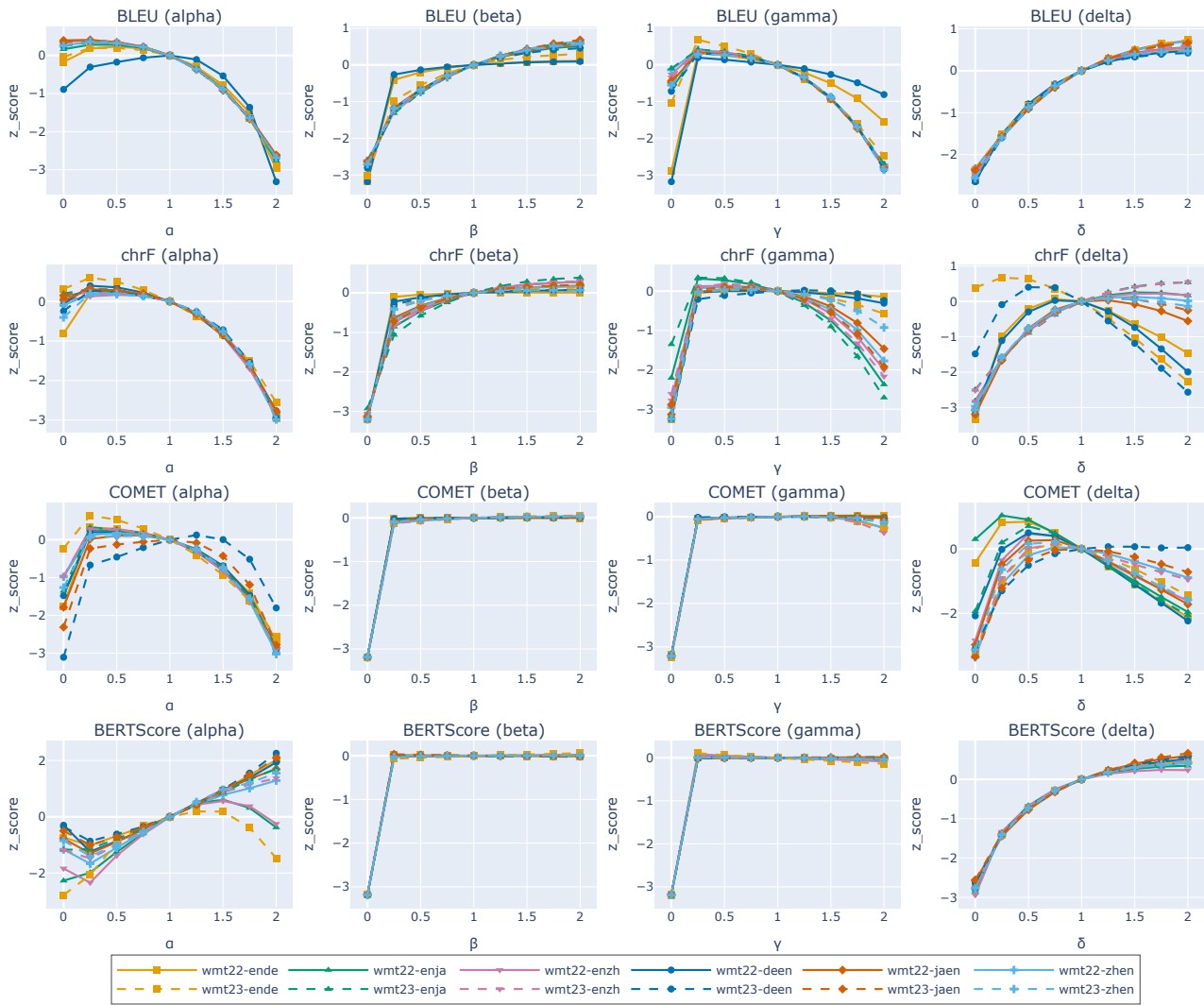

*Figure 1.* Results on machine translation tasks across 12 evaluation scenarios spanning language directions and datasets. Rows correspond to evaluation metrics, and columns correspond to the term weights $\alpha$, $\beta$, $\gamma$, and $\delta$. Each plot reports the result obtained by grid searching over $[0.00, 0.25, 0.50, 0.75, 1.00, 1.25, 1.50, 1.75, 2.00]$ for the corresponding term weight. We report z-scores normalized with respect to the configuration where the corresponding term weight is set to 1.0, facilitating a unified interpretation and analysis.

ing channel weight is set to 1.0.

**Finding 1 (Overview): Channel-wise trends characterize individual utility functions, while the overall MBR behavior follows an almost consistent pattern.** Figure 1 shows that although there is variation across tasks and languages pairs, the overall shapes of the curves are largely consistent within each evaluation metric, tracing similar trajectories across term weights. Furthermore, we observe several characteristic patterns, including consistently monotonic increases, e.g., $\delta$ for BLEU and BERTScore, monotonic decreases, e.g., $\alpha$ for BLEU and chrF, and relatively stable behavior, e.g., $\beta$ and $\gamma$ for COMET and BERTScore. These differences reflect the distinct aspects emphasized by each utility function, indicating that our analysis successfully

captures metric-specific characteristics. In the context of MBR decoding, this suggests that different utility functions prioritize different criteria, motivating the use of diverse evaluation metrics depending on the desired generation behavior. From a stability perspective, BLEU and COMET generally exhibit similar trends across tasks. While COMET and chrF show task-dependent sensitivity to changes in $\delta$ under some settings, the global trend remains consistent. For BERTScore, we observe task- and language-pair-dependent differences in trends with respect to $\alpha$ at higher weight values; however, conservative parameter ranges, e.g., values up to 1.5, yield consistent behavior across tasks. Overall, the noisy-channel-based decomposition enables a fine-grained analysis of utility functions by examining term-wise weight variations, going beyond prior empirical evaluations that

relied solely on final score comparisons. While we identify metric-specific tendencies and characteristic features, we also find that the overall effect of MBR decoding follows broadly similar trajectories across utility functions.

**Finding 2 ($\alpha$): The hypothesis-to-reference likelihood $P(r_j \mid h_i)$ shows monotonic effects with respect to its weight and generally prefers low weighting, except in the case of BERTScore.** Varying the weight $\alpha$ reveals that BLEU and chrF exhibit clear monotonic decreases as $\alpha$ increases. COMET shows a similar decreasing trend for $\alpha > 0.25$. In contrast, BERTScore displays the opposite behavior, exhibiting a monotonic increase with respect to $\alpha$. Although performance degradation can occur at very high values in some settings, e.g., WMT'23 En→De, BERTScore consistently improves up to moderately high weights such as $\alpha = 1.25$ or $1.5$. These results indicate that the hypothesis-to-reference likelihood $P(r_j \mid h_i)$ is generally more effective when assigned a low weight, as overemphasizing this term can lead to performance degradation for most metrics. Notably, BERTScore constitutes a clear exception: for many tasks, assigning a high weight, e.g., $\alpha = 2.0$, yields substantial improvements. We attribute this behavior to the design of BERTScore, which computes an F-score based on symmetric token-level alignment between hypotheses and references. As a result, explicitly emphasizing hypothesis-to-reference alignment aligns well with BERTScore's underlying assumptions. Overall, these findings demonstrate that the hypothesis-to-reference likelihood is a major factor influencing MBR decoding performance. While low weights are generally preferable, certain symmetric, alignment-based metrics such as BERTScore benefit from higher weights. This metric-dependent effect constitutes a previously hidden factor revealed by the noisy-channel-based decomposition.

**Finding 3 ($\beta$): The hypothesis prior $P(h_i)$ has limited impact for semantic-aware metrics such as COMET, but provides modest gains for surface-level metrics such as BLEU by acting as a semantic correctness term.** Figure 1 shows that varying $\beta$ generally does not lead to large performance changes. In most cases, COMET and BERTScore exhibit stable behavior for any non-zero value of $\beta$. In contrast, for BLEU and chrF, slightly larger values of $\beta$ ($\beta > 1$) yield modest improvements over original MBR decoding, or at least do not degrade performance. Overall, changes in the weight of the hypothesis prior $P(h_i)$ exhibit consistent behavior across settings. Down-weighting $\beta$ does not lead to degeneration for neural metrics such as COMET and BERTScore, whereas for surface-level matching metrics like BLEU and chrF, under-weighting can result in performance degradation for certain tasks. Conversely, over-weighting $\beta$ can yield small but consistent gains for some utility functions, producing results that are comparable to

or better than original MBR decoding. These observations suggest that while $P(h_i)$ has limited impact when using metrics that explicitly account for sentence-level semantics, e.g., BERTScore, it plays a more important role for surface-level metrics that lack semantic modeling, e.g., BLEU. By incorporating the prior probability of hypotheses, the model implicitly captures plausibility among candidate sentences, leading to improved scores for metrics that rely on surface-form alignment. Consequently, the hypothesis prior $P(h_i)$ is particularly beneficial when the utility function does not incorporate semantic understanding.

**Finding 4 ($\gamma$): The reference-to-hypothesis likelihood $P(h_i \mid r_j)$ is most effective with moderate low weights, while high weights can lead to performance degradation.** For the reference-to-hypothesis likelihood $P(h_i \mid r_j)$, we observe that setting $\gamma > 1$ leads to performance degradation for most metrics except BERTScore, with particularly pronounced drops for BLEU and chrF in some language pairs and tasks. Accordingly, this suggests that at higher weights, this term does not provide measurable benefits across metrics. In contrast, moderately low weights, e.g., $\gamma = 0.25$ or $0.5$, either avoid degradation or yield modest gains for surface-level metrics such as BLEU and chrF in some cases. These results indicate that overemphasizing the reference-to-hypothesis direction can be detrimental to performance. Notably, original MBR decoding implicitly relies on $P(h_i \mid r_j)$, as shown in Table 1. Our findings suggest that down-weighting this term and instead emphasizing complementary terms leads to more robust performance. The proposed noisy-channel-based decomposition makes this trade-off explicit and highlights the importance of balancing term contributions.

**Finding 5 ($\delta$): The reference prior $P(r_j)$ dynamically influences performance, exhibiting metric-dependent stability while responding monotonically to its weight.** We observe that the reference prior weight $\delta$ exhibits clear monotonic effects for BLEU and BERTScore, where increasing $\delta$ leads to smooth and consistent performance improvements. This indicates that the reference prior $P(r_j)$ plays an important role for these metrics. In contrast, chrF and COMET exhibit more varied behaviors: performance may peak around $\delta = 1$, follow a parabolic trajectory, or even prefer lower weights in some settings. These observations suggest that the quality of the reference prior can substantially affect MBR decoding performance, although its impact depends on the evaluation metric. Notably, as discussed in Sections 2 and 3, model-based MBR decoding (Jinnai et al., 2024) explicitly incorporates model output probabilities as the reference prior $P(r_j)$. Our analysis shows that this design choice is particularly effective for metrics such as BLEU and BERTScore, while its impact is less stable for chrF and COMET. Consistent with Jinnai

et al. (2024), which primarily reports gains on BLEU and BERTScore rather than chrF or COMET, our results provide a theoretical follow-up that explains these empirical findings. Specifically, our channel-wise analysis clarifies the conditions under which modeling for prior $P(r_j)$ is most effective. For utility functions that exhibit monotonic gains with respect to $\delta$, improving the quality of the reference prior and assigning it a higher weight is desirable.

### 4.1.3. SUMMARY: RESEARCH ANSWERS FOR RQ1

Our analysis reveals that while utility functions exhibit metric-specific characteristics, the hypothesis-to-reference likelihood $P(r_j \mid h_i)$ and the reference prior $P(r_j)$ are the most sensitive factors influencing performance. In contrast, the hypothesis prior $P(h_i)$ has a limited impact across utility functions, and the reference-to-hypothesis likelihood $P(h_i \mid r_j)$ generally benefits from conservative weighting.

Overall, these results indicate that although the global behavior of MBR decoding admits a unified interpretation, utility-specific differences become evident through term-wise weighting. Consequently, it is important to carefully consider the choice of utility function in MBR decoding. While conventional metrics offer limited flexibility for fine-grained adjustment, the proposed noisy-channel-based decomposition enables explicit control over individual components. By adjusting term weights, our framework may yield performance gains for each utility function.

### 4.2. RQ2: Task-Agnosticity under Utility Functions

#### 4.2.1. SETTINGS

To examine whether channel importance is task-agnostic, we analyze whether consistent trends emerge across different tasks when the utility function is fixed. We additionally conduct experiments on summarization and captioning tasks.

Following Jinnai et al. (2024); Kamigaito et al. (2025); Deguchi & Nagata (2025), we examine three summarization datasets: CNN/DailyMail (CNN/DM) (Nallapati et al., 2016), XSum (Narayan et al., 2018), and SAMSum (Gliwa et al., 2019). For summary generation, we employ BART-based fine-tuned models (Lewis et al., 2020): `facebook/bart-large-cnn`, `facebook/bart-large-xsum`, and `philschmid/bart-large-cnn-samsum` for CNN/DM, XSum, and SAMSum, respectively.

For caption generation, we test on two benchmarks: MSCOCO (Lin et al., 2014) using the split of Karpathy & Fei-Fei (2015), and NoCaps (Agrawal et al., 2019). We employ `Salesforce/blip2-flan-t5-xl-coco` and `Salesforce/blip2-flan-t5-xl` (Li et al., 2023) for generation on MSCOCO and NoCaps, respectively.

We use BERTScore (Zhang et al., 2020) with pretrained

model `microsoft/deberta-xlarge-mnli` (He et al., 2021) as both the utility function and evaluation metric. We employ HuggingFace Transformers (Wolf et al., 2020) for candidate sampling, and all remaining settings follow those described in Section 4.1.1.

#### 4.2.2. RESULTS

Figure 2 shows the results. To facilitate cross-task comparison, we also include translation results from WMT'22 and WMT'23 for the En↔De language pair, as reported in Section 4.1.2. Since only a limited number of evaluation metrics are consistently applicable and reliable across tasks, we follow Jinnai et al. (2024); Kamigaito et al. (2025) and use BERTScore as a unified cross-task evaluation metric.

**Finding 6: The overall trajectory remains consistent across tasks and datasets, indicating that MBR decoding behavior is task-agnostic and primarily dominated by the choice of utility function.** Figure 2 shows that similar trajectories are observed across different tasks. In particular, the trends for $\beta$, $\gamma$, and $\delta$ are largely consistent across benchmarks. In contrast, variations in $\alpha$ can lead to performance degradation in some benchmarks when $\alpha > 1.5$; however, within the range $0.25 \leq \alpha \leq 1.5$, performance remains comparable across tasks. Overall, the trajectories are smooth and exhibit continuous changes with respect to term weights. These results indicate that, as long as extreme parameter values are avoided, a fixed, stable utility function, i.e., BERTScore, exhibits consistent behavior across tasks and datasets.

**Finding 7: Optimal channel weights are largely task-independent, and for BERTScore, the hypothesis-to-reference likelihood $P(r_j \mid h_i)$ and the reference prior $P(r_j)$ dominate performance.** Figure 2 shows that performance is largely insensitive to the weights of the hypothesis prior $P(h_i)$ and the reference-to-hypothesis likelihood $P(h_i \mid r_j)$, indicating that these terms are not critical factors across tasks. In contrast, increasing the weights of the hypothesis-to-reference likelihood $P(r_j \mid h_i)$ and the reference prior $P(r_j)$ consistently leads to performance gains. Notably, assigning a moderately high weight to $P(r_j \mid h_i)$ together with a high weight to $P(r_j)$ yields robust improvements across tasks. These findings also provide insight into why model-based MBR decoding (Jinnai et al., 2024), which introduces additional weighting terms, performs well in practice. In particular, the presence of a consistent reference prior $P(r_j)$ across tasks plays a crucial role. Our analysis highlights the importance of explicitly modeling this term, and BERTScore exhibits stable, clear monotonicity with respect to $P(r_j)$, leading to improved performance across diverse tasks. By contrast, follow-up experiments (Deguchi et al., 2024a) on model-based MBR

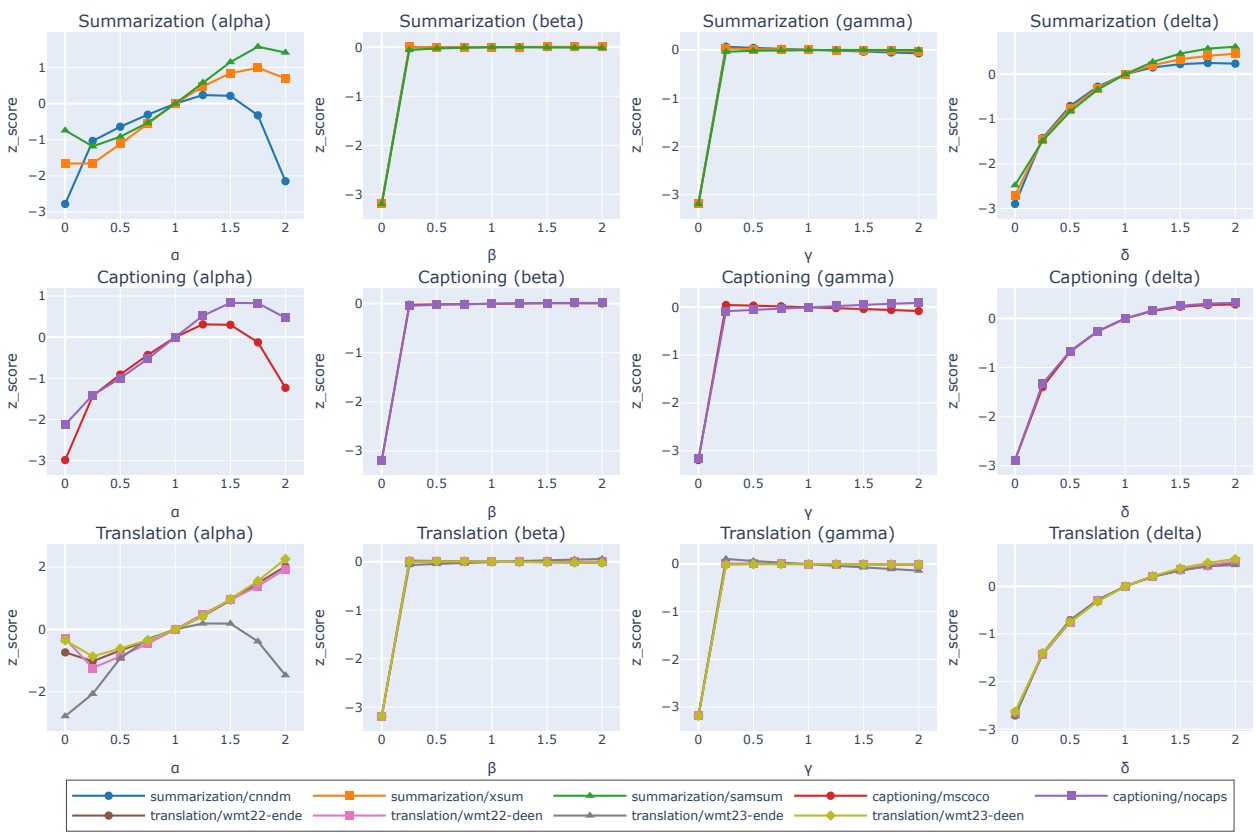

*Figure 2.* Results on summarization, captioning, and machine translation tasks with BERTScore. The machine translation results are taken from Figure 1 to facilitate cross-task comparison. Rows correspond to tasks, and columns correspond to the term weights $\alpha$, $\beta$, $\gamma$, and $\delta$. Each plot reports results obtained by grid searching over $[0.00, 0.25, 0.50, 0.75, 1.00, 1.25, 1.50, 1.75, 2.00]$ for the corresponding term weight. We report z-scores normalized with respect to the configuration where the corresponding term weight is set to 1.0.

showed that Monte Carlo-based approaches, i.e., original MBR, often performed better. We attribute this discrepancy to differences in utility function selection, as BERTScore was not included in their evaluation. As shown in Figure 1, the other metrics, such as BLEU, are influenced not only by the reference prior $P(r_j)$ but also by the reference-to-hypothesis likelihood $P(h_i \mid r_j)$. In contrast, BERTScore is largely dominated by the reference prior $P(r_j)$, making it particularly well aligned with model-based MBR decoding. By explicitly accounting for the choice of utility function, our analysis provides a unified interpretation that bridges these seemingly conflicting experimental findings.

**Finding 8: The observed trends are consistent across tasks even when using another metric, i.e., BLEU, suggesting that the findings are generalizable.** We further investigated the generalizability of our findings by analyzing the trends observed in image captioning using BLEU. Figure 3 presents the cross-task trajectories under BLEU. The results show that the trajectories are largely consistent across tasks. While the effects of parameter variations differ depending on the characteristics of each model, the overall

trends remain stable across tasks. These observations provide additional evidence supporting our previous findings and suggest that the identified trends are not specific to a particular task or dataset.

### 4.2.3. SUMMARY: RESEARCH ANSWERS FOR RQ2

We investigate whether term-wise importance is task-agnostic by fixing BERTScore as the utility function, one of the few well-established metrics applicable across tasks. Despite differences in task type, we observe consistent trends. This indicates that MBR decoding performance is more influenced by the choice and characteristics of the utility function than by the task type. Furthermore, as shown in Figures 1 and 2, several plots exhibit parabolic-like variations with smooth and continuous trajectories. This suggests that performance with respect to term weights typically peaks at a certain point and then changes direction in a parabolic-like manner beyond that optimum. Our noisy-channel-based decomposition revealed that these optimal points reflect metric- and task-specific characteristics, enabling a principled and interpretable understanding of how term weighting affects

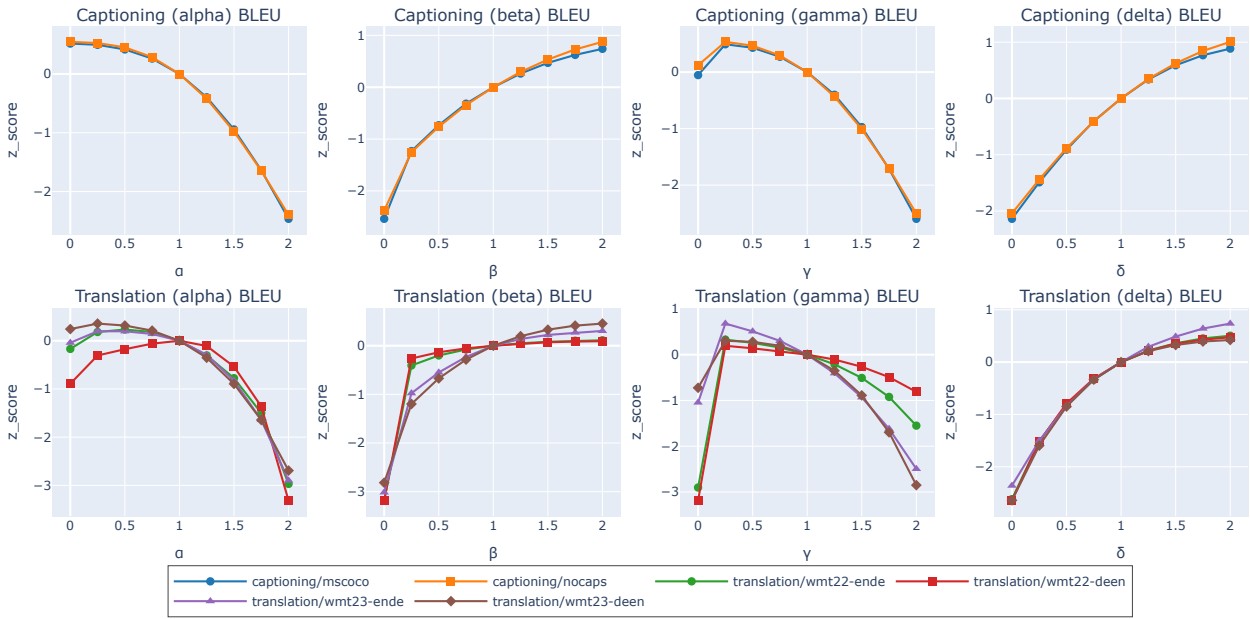

*Figure 3.* Results on captioning and machine translation tasks with BLEU. The machine translation results are taken from Figure 1 to facilitate cross-task comparison. Rows correspond to tasks, and columns correspond to the term weights $\alpha$, $\beta$, $\gamma$, and $\delta$. Each plot reports results obtained by grid searching over $[0.00, 0.25, 0.50, 0.75, 1.00, 1.25, 1.50, 1.75, 2.00]$ for the corresponding term weight. We report z-scores normalized with respect to the configuration where the corresponding term weight is set to 1.0. Similar to the BERTScore results shown in Figure 2, the trajectories exhibit similar trends across different tasks.

MBR decoding performance.

## 5. Conclusion

In this study, we introduced a noisy-channel-based decomposition of MBR decoding, which reformulates MBR scoring into four distinct probabilistic components: the hypothesis-to-reference likelihood $P(r_j \mid h_i)$, the hypothesis prior $P(h_i)$, the reference-to-hypothesis likelihood $P(h_i \mid r_j)$, and the reference prior $P(r_j)$. Inspired by the source channel model, we assign explicit weights to each component, enabling a unified interpretation and improved interpretability by isolating the contribution of each channel.

In our empirical analyses, we investigated two key questions: how the choice of utility function affects MBR decoding performance, and whether term-wise importance exhibits consistent trends across tasks. Our results show that while channel-wise behaviors are largely task-agnostic and follow similar global trajectories, performance differences are primarily driven by the characteristics of the utility function. Furthermore, our channel-wise analysis provides analytical support for the effectiveness of the reference prior $P(r_j)$ in model-based MBR decoding. The proposed decomposition reveals that finer-grained weighting of this term offers additional room for performance gains. Beyond the reference prior, our analysis also uncovers the previously underexplored importance of the hypothesis-to-reference likelihood

$P(r_j \mid h_i)$, demonstrating that it contributes meaningfully to MBR decoding performance despite often being treated implicitly in prior work.

Although the primary contribution of this work is to provide a new perspective on the mechanisms underlying MBR decoding, our analysis also suggests that reweighting can yield performance improvements over standard MBR. For example, when applying the best weights found on WMT22 En–De to WMT23 En–De, performance improves from 50.05 (standard MBR; $(\alpha, \beta, \gamma, \delta) = (0, 0, 1, 1)$ in Table 1) to 50.18 on ChrF, demonstrating that the benefits of reweighting transfer across settings. Combined with recent advances in MBR decoding discussed in Section 2, these findings suggest that MBR can be further leveraged to achieve direct performance improvements. Notably, the effectiveness of reweighting appears to depend more on the characteristics of the metric than on a particular dataset or task. This observation points to promising new directions for MBR decoding, such as developing improved utility functions from a source-channel perspective.

To summarize, our findings highlight the importance of the reference prior and the hypothesis-to-reference likelihood in MBR decoding, while demonstrating that the suitability of a utility function can be systematically analyzed through term-wise behavior. More broadly, our proposed decomposition provides a principled and interpretable framework for understanding, analyzing, and improving MBR decoding.

## Impact Statement

This work examines the interpretability of MBR decoding through a noisy-channel-based decomposition, with the goal of advancing machine learning. All tools, models, and datasets used in this study are publicly available and permitted for research purposes. In particular, the MBRS library (Deguchi et al., 2024a) used in our experiments is released under the MIT License, which allows modification and reuse. Therefore, we confirm that this work is fully compliant with all applicable licenses.

For writing assistance, we used conventional translation and grammar-checking tools such as DeepL and Grammarly. All aspects of this research, including the research ideas, technical content, experimental design, analysis, and manuscript writing, were carried out entirely by the authors under responsible authorship. We did not intentionally use generative AI systems that provide content suggestions, and we affirm that their use was limited to language refinement.

Regarding code release, we are currently preparing a repository that consolidates a series of our recent studies on MBR decoding, including the method presented in this paper. The repository will be made publicly available upon completion. In the meantime, if early access is needed for verification or reproducibility purposes, please feel free to contact us.

Finally, to avoid hallucinated or incorrect citations, as highlighted by Sakai et al. (2026), all references were cited from and verified against reliable original sources by the authors. Wherever possible, we provide direct and verifiable citation information, such as clickable links or DOIs. We also added several citations in the camera-ready version (Freitag et al., 2024; Yu et al., 2017; Yang et al., 2024; Amrhein & Sennrich, 2022; Finkelstein et al., 2024; Soetedjo et al., 2026). In addition, Vashurin et al. (2025) was flagged by the automated reference checker provided by ICML. However, this paper was published at NeurIPS 2025, and we have verified the citation and its associated link. The issue appears to stem from the paper not yet being indexed by services such as DBLP or Crossref. Therefore, we believe this is a false positive generated by the automated checker.

Based on these considerations, we believe that this paper does not present any specific ethical, legal, scientific, or societal risks that require additional discussion.

## Acknowledgments

We thank the anonymous reviewers for their valuable comments and suggestions. Their feedback has strengthened this work, encouraged its publication, inspired new research directions, and motivated us. We also sincerely thank the Area Chairs, Senior Area Chairs, and Program Chairs for facilitating a smooth review process and providing us with the opportunity to present this work.

We are grateful to Katsuhiko Hayashi for inspiring Hidetaka Kamigaito to invent the decomposition (Section 3.1) and its interpretation (Section 3.2) through discussions on the normalization of utility functions in their previous work (Kamigaito et al., 2025). We also thank Hiroyuki Deguchi for valuable discussions on the foundations of MBR decoding (Section 2), which provided mathematical insights and deepened our understanding of the underlying theory.

This work was supported by JSPS KAKENHI Grant Numbers JP23K28148, JP25K24369, and JP26K21312.

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
