# OpenReview forum: "Noisy-Channel Minimum Bayes Risk Decoding"
_ICML.cc/2026/Conference — ICML 2026 regular_

### Official Review · Reviewer_zqrt · 2026-03-10

**Soundness:** 2
**Presentation:** 4
**Significance:** 3
**Originality:** 3
**Overall Recommendation:** 4
**Confidence:** 4

**Summary:**

In this paper, the authors propose a noisy-channel reinterpretation of Minimum Bayes Risk (MBR) decoding by decomposing the scoring function into four weighted components: hypothesis likelihood P(r|h), reverse likelihood P(h|r), hypothesis prior P(h), and reference prior P(r). The authors argue that this formulation unifies several existing MBR variants and enables explicit handling of asymmetric utilities. Further, the authors highlight that a weighting of the individual factors can yield a positive impact in the results. The paper analyses weight sweeps in [0, 2] across multiple tasks (MT, summarization, captioning) and metrics (BLEU, chrF, COMET, BERTScore), reporting qualitative performance trends.

**Compliance With Llm Reviewing Policy:**

Affirmed.

**Final Justification:**

The authors provided a good and thorough clarification of my main concerns, which were mostly related to the soundness and methodological correctness of the method. These concerns have been partially resolved regarding soundness and fully resolved regarding methodological clarity. If the authors incorporate these improvements in the final revision, the paper's clarity can substantially improve, and I believe it has the potential to be a good contribution to the field.

**Key Questions For Authors:**

1. As mentioned before, how do you ensure that utilities like COMET (with potentially negative outputs) yield valid, nonnegative quantities for normalization in Eqs. (7), (9), and (13)? Do you apply shifts, clipping, or some other transforms?
2. Eq. (13) normalizes P(rj|hi) over hypotheses for a fixed rj rather than over references for a fixed hi. Is this a typo, or do you intend a “channel-like” score rather than a true conditional? If the latter, would Bayes ’ rule no longer apply?
3. How are the final channel weights selected in practice? Do you use a dev set and fix weights per metric/task, or do you propose a metric-agnostic default?
4. I could not see the computational overhead of your method mentioned. How is this extension relative to the original MBR for your best-performing configurations?
5. An analysis of weighting vs. human preferences would be interesting, I assume, based on previous experience, that for many tasks, it will not be monotonic. Finding patterns here could be very insightful.

**Limitations:**

No. The paper contains a brief impact statement claiming that the work does not raise ethical or societal risks, but it does not include a discussion of methodological limitations. It would strengthen the paper if the authors explicitly discussed limitations such as the dependence on automatic evaluation metrics, the lack of cross-metric or human evaluation, and the potential sensitivity of the method to model architecture or sampling procedures.

**Strengths And Weaknesses:**

## Strengths

1. I found the paper well-written, well-structured, and easy to follow. Overall, the presentation is good.

2. As far as I know, the channel-based decomposition is novel, and it offers, at the same time, an intuitive lens to analyze directional effects of asymmetric evaluation metrics.

3. The authors provide detailed experiments spanning multiple tasks and datasets. In particular, they test the sensitivity of the utilities with respect to hyperparameter sweeps.

4. This approach provides insights into the role of the reference prior and the hypothesis-to-reference likelihood, expanding the typical view on MBR decoding.

5. I believe the framework could help practitioners better understand and tune MBR decoding.

6. The paper addresses a relevant problem that remains challenging: how to select outputs such that the quality of generated text is improved across multiple tasks. The proposed framework offers opportunities for further interpretations of MBR.

---

## Weaknesses

1. As far as I can see, the authors did not provide a link to an anonymous GitHub repository. This makes it difficult to check the implementation details and verify the reported results, which affects reproducibility.

2. Moving to the formulas presented in the paper: if I understand correctly, the operational definition of \(P(r|h)\) (Eq. 13) normalizes over hypotheses rather than references. In that case, it would not define a valid conditional distribution, which would appear inconsistent with the Bayesian decomposition used to justify the noisy-channel interpretation.

3. Metrics such as COMET can produce negative scores. The paper does not explain how utilities are transformed to satisfy the non-negativity assumptions required for normalization. Please clarify this in the Methodology or Appendix.

4. Higher values in automatic metrics are not necessarily a symptom of higher output quality, but may reflect overfitting to references. In my opinion, this study lacks stronger baselines for assessing quality beyond automatic metrics that, depending on the task, can be more or less reliable for assessing output quality. Even if imperfect, an LLM-as-a-judge (with an external or "impartial" judge) could provide additional insights. Alternatively, one could analyze how the method ranks human-produced sequences. Relatedly, it would be interesting to know what channel weighting empirically works best across tasks when evaluated against human preferences.

5. If Bayes’ rule holds, \(P(h|r)P(r)=P(r|h)P(h)\), then the four-term product does not appear to constitute a new probabilistic decomposition unless the terms are independently modeled. Could the authors clarify this point?

6. The authors do not report confidence intervals or test the statistical significance of their results. How strong is the variability in the reported improvements? In my opinion, reporting uncertainty estimates should be standard practice in this type of empirical studies.

7. Information about architecture-related effects is not presented, but instead the model architecture appears to be fixed. I understand this given the focus of the paper, but I wonder whether the conclusions would transfer to settings with different models.

---

> ### Author Rebuttal · Authors · 2026-03-31
>
> Dear zqrt,
>
> We would like to thank you for the valuable feedback and for recognizing many strengths of our work.
>
> ---
> **Weaknesses:**
> > W1: GitHub Link
>
>  We will include the GitHub link in the final version.
>
> (During the rebuttal phase, external links are restricted to figures/tables and their captions.)
>
> ---
> > W2: Regarding Eq. (13)
>
> In our formulation, we define inversion as swapping f(h_i, r_j) to f(r_j, h_i). Following the same derivation as in Eqs. (7), (8), and (9), we can analogously derive P(r_j \mid h_i). We will provide a more detailed explanation of this transformation in the final version.
>
> ---
> > W3: Treatment of negative utility values
>
> Since Eq. (6) takes an argmax, the objective is invariant under additive shifts. Thus, if utility values include negative numbers, we can shift them so that the minimum becomes zero without affecting the result. In addition, the COMET model we use (wmt22-comet-da) outputs scores in the [0,1] range and does not produce negative values, which we have verified. We will explicitly clarify this in the revision.
>
> ---
> > W4: Human preference and LLM-as-a-judge
>
> Automatic metrics are typically evaluated based on their correlation with human preferences through meta-evaluation studies. In this sense, they already provide a reasonably reliable proxy for human judgment. Moreover, LLM-as-a-judge approaches also require validation against human preferences, making their reliability non-trivial to establish.
>
> Our work focuses on providing a fundamental and theoretical analysis of MBR decoding, along with a comprehensive empirical study that systematically sweeps utility functions. Since the metrics we use are known to correlate well with human preferences, we believe the concern is already reasonably addressed within our experimental framework. We will further clarify this point in the final version.
>
> ---
> > W5: The four-term product does not appear to constitute a new probabilistic decomposition unless the terms are independently modeled.
>
> As described in L147, we model these terms within a source-channel framework. Source-channel models apply weighting to probabilistic components. We will add a clearer explanation of the source-channel model interpretation in the revision.
>
> ---
> > W6: How strong is the variability in the reported improvements?
>
> Thank you for the comment. We report z-score normalized results, which standardize the variance across plots, resulting in a variance of 1 in each case. This normalization enables fair comparison across different metrics and tasks, facilitating a comprehensive trend analysis. As a result, each trajectory reflects normalized values, providing a statistically valid basis for comparison. Nevertheless, we will improve the plots by enhancing clarity and, where possible, adding confidence bands. Thank you for the helpful suggestion.
>
> ---
> > W7: Information about model architecture-related effects
>
> The models used for generation differ across tasks, following common practice in prior work. As you acknowledged, the focus of this paper is on the theoretical derivation and comprehensive analysis of utility functions. We will clarify this point and discuss architecture-related effects as a limitation and direction for future work.
>
> ---
> **Key Questions:**
> > Q1: Treatment of negative utility values
>
> Please see our response to W3. Since the objective involves an argmax, negative values can be handled via shifting without affecting the result. In practice, negative values did not occur in our experiments. We will clarify this in the revision.
>
> ---
> > Q2: Regarding Eq. (13)
>
> Please see our response to W2. The derivation follows analogously from Eq. (7) to Eq. (8).
>
> ---
> > Q3: How are the final channel weights selected in practice?
>
> While weight sensitivities vary across metrics, they remain stable across tasks. Therefore, in practice, a fixed set can be applied. For example, when applying the best weights found on WMT22 En–De to WMT23 En–De, we observe an improvement from 50.05 (standard MBR) to 50.18, demonstrating that our reweighting yields consistent gains across settings. We will clarify this point in the final version.
>
> ---
> > Q4: Computational overhead
>
> Our proposed decomposition mainly involves simple operations such as weighted sums and scaling, and can be computed efficiently in a sequential manner. As a result, it introduces negligible overhead compared to standard MBR. We will clarify this in the revision.
>
> ---
> > Q5: An analysis of weighting vs. human preferences
>
> Please see our response to W4. The metrics used in our analysis are known to correlate well with human preferences, and we conduct a comprehensive sweep over all weighting patterns. The resulting insights are summarized in Findings 1-7. We will further strengthen the discussion in the revision, particularly regarding practical guidance for weight selection.
>
> ---
> > Limitations
>
> Thank you for the suggestion. We will explicitly add a Limitations section and expand the discussion.
>
> Best,

---

> > ### Author Rebuttal · Reviewer_zqrt · 2026-04-02
> >
> > Dear authors,
> >
> > Thank you for your response. I appreciate the effort you put into clarifying several aspects of the paper, particularly regarding the treatment of negative utilities, computational overhead, and practical considerations around weight selection.
> >
> > That said, some of my main concerns remain insufficiently addressed. In particular, the definition of
> > P(r∣h) in Eq. (13) does not appear to correspond to a valid conditional distribution, as it is normalized over hypotheses rather than references. In my opinion, this weakens the probabilistic interpretation underlying the proposed decomposition.
> >
> > Relatedly, the interpretation of the four-term product as a probabilistic (Bayesian) decomposition remains conceptually unclear. Since the components are derived from the same utility function rather than modeled independently, it is difficult to view this as a true probabilistic factorization rather than a reweighting scheme.
> >
> > On the empirical side, the absence of uncertainty estimates (e.g., confidence intervals or significance testing) and the reliance solely on automatic metrics limit the strength of the conclusions. In particular, the concern regarding potential overfitting to metrics and the lack of validation against human preferences or independent evaluation signals remains open.
> >
> > I appreciate the paper's strengths, including its clarity of presentation, the experiments, and the interesting perspective on MBR decoding. I believe the work has potential, but addressing the points above would be important for strengthening both the theoretical grounding and the empirical validation.
> >
> > As a result, I am keeping my original assessment.
> >
> > Yours sincerely,
> >
> > Reviewer zqrt

---

> > > ### Author Response · Authors · 2026-04-06
> > >
> > > Dear Reviewer zqrt,
> > >
> > > We are glad that some of your concerns were addressed in our initial response, and we sincerely appreciate your recognition of the strengths of our work, e.g., *“clarity of presentation, the experiments, and the interesting perspective on MBR decoding”.* Due to space limitations, some of our responses may have been brief or insufficient. In this final rebuttal, we focus on your remaining **three concerns**.
> > >
> > > ---
> > > > 1: The definition of P(r∣h) in Eq. (13) does not appear to correspond to a valid conditional distribution
> > >
> > > Thank you for pointing this out. Upon careful checking, we found that this issue is due to a typo, as you expected: the normalization should be taken over references for a fixed $h_i$, i.e., $\sum_{r_{j'} \in \mathcal{R}} f_\theta(r_{j'}, h_i)$. The current expression mistakenly reflects an incorrect summation due to a copy error from Eq. (8).
> > >
> > > As the derivation of $P(r_j \mid h_i)$ is obtained analogously to $P(h_i \mid r_j)$, the intended formulation is symmetric and consistent. Therefore, while our initial response was not incorrect, it did not fully capture your intention. We apologize for any confusion this may have caused, including to you, ACs, and other reviewers.
> > >
> > > However, importantly, this typo does not affect the overall correctness of the derivation. The derivation, implementation, and experiments all follow the correct formulation. Eq. (13) serves as a supplementary and independent expression and is included only for explanatory purposes. Therefore, this can be addressed with minor corrections without affecting the overall conclusions. We will correct this typo and unify the notation in the final version.
> > >
> > > ---
> > > > 2: The interpretation of the four-term product as a probabilistic (Bayesian) decomposition remains conceptually unclear.
> > >
> > > Our formulation is based on the concept of the noisy-channel / source-channel model. We first derive the decomposition from a noisy-channel perspective and then introduce component weighting, a standard approach in source-channel modeling [1, 2]. This formulation enables us to control the importance of each channel.
> > >
> > > Up to Eq. (12), the formulation is a probability-based reformulation. In Eq. (14), we expand the decomposition in Eq. (12) to a weighted form based on the concept of the source-channel model. Since the objective of MBR decoding is defined via an argmax, this transformation should be understood as a decision-theoretic expansion rather than a strict probabilistic factorization.
> > >
> > > To clarify this point, we will change the wording, e.g., replacing “reformulate” with “expand” in L150, and explicitly state that this is not a true probabilistic decomposition but a transformation of the decision objective in Eq. (14). This will clarify where probabilistic and decision-theoretic concepts are used, and we hope it will make the paper easier to understand.
> > >
> > > [1] Discriminative Training and Maximum Entropy Models for Statistical Machine Translation (Och & Ney, ACL 2002). (p.3, Fig. 2, around Eq. 8, is intuitive)
> > >
> > > [2] The Neural Noisy Channel (Yu et al., ICLR 2017). arXiv:1611.02554. (Section 3.1)
> > >
> > > ---
> > > > 3: The absence of uncertainty estimates and reliance on automatic metrics limit the strength of the conclusions.
> > >
> > > We would like to clarify this point. The fundamental objective of MBR decoding is to select candidates that maximize a given utility function, grounded in a decision rule. In the context of MBR decoding, the goal is achieved by using a utility function that correlates well with human preferences, and we use such highly correlated metrics in our experiments [3].
> > > From this perspective, we respectfully note that concerns such as overfitting or human preference evaluation are not aligned with this objective, and we believe it would not yield meaningful implications. On this point, we ask that our position be acknowledged. We will add this clarification in the revision.
> > >
> > > Similarly, we prioritize capturing global trends and trajectories. These trajectories are smooth trends derived from thousands of points and capture global behavior while accounting for variability through z-scores. Therefore, we believe such testing would provide similar implications and trends, as the trajectories are smooth and consistent.
> > >
> > > Finally, our experimental design and protocol follow common practices in MBR. We believe our evaluation is appropriate and that our extensive parameter sweep provides meaningful empirical insights, accounting for variance. We kindly ask that you acknowledge our contribution at this point. We will explain the experimental design in more detail in the revision.
> > >
> > > [3] Are LLMs Breaking MT Metrics? Results of the WMT24 Metrics Shared Task (Freitag et al., WMT 2024)
> > >
> > > ---
> > > Thank you again for your valuable feedback, which has helped strengthen the manuscript. We believe these concerns can be addressed with minor revisions in the final version. We kindly ask you to reconsider your evaluation or provide further feedback.
> > >
> > > Best,

---

### Official Review · Reviewer_6oCu · 2026-03-12

**Soundness:** 2
**Presentation:** 2
**Significance:** 2
**Originality:** 3
**Overall Recommendation:** 3
**Confidence:** 3

**Summary:**

This paper studies Minimum Bayes Risk (MBR) decoding for text generation, motivated by the fact that many widely used utility functions and evaluation metrics are asymmetric in their hypothesis–reference direction. The authors propose a noisy-channel-inspired reformulation that factors MBR scoring into four components meant to capture bidirectional interactions and priors: a hypothesis-to-reference term, a reference-to-hypothesis term, a hypothesis prior, and a reference prior. Building on this factorization, they introduce per-channel weights and use the resulting framework to (i) give a unifying interpretation of several existing MBR variants and (ii) analyze how channel weights interact with different utility functions and tasks. The empirical study sweeps channel weights on machine translation, summarization, and captioning, and reports metric-specific and largely task-stable trends, especially when using BERTScore as the utility.

**Compliance With Llm Reviewing Policy:**

Affirmed.

**Final Justification:**

The authors addressed my questions with clarity and evidence. I have increased my score.

**Key Questions For Authors:**

1. Please state explicit definitions for \(P(h_i \mid r_j)\), \(P(r_j)\), \(P(h_i)\), and \(P(r_j \mid h_i)\), including how you transform raw utility scores into non-negative values suitable for normalization. If a metric can produce negative values, what exact handling do you use?
2. In the derivation, can you justify the step from Eq. (9) to Eq. (10) as written? If there is a missing term (for example a power on \(P(r_j)\) or a square root applied outside the sum), please correct it and explain why the resulting objective is equivalent to standard MBR.
3. In Eq. (13), should the normalization for \(P(r_j \mid h_i)\) sum over references (varying \(r\) for fixed \(h_i\)) rather than over hypotheses (varying \(h\) for fixed \(r_j\))? If the current equation is intentional, what is the probabilistic meaning of that quantity?
4. What is the exact selection protocol for channel weights? Are weights chosen on a held-out development set, or are the same evaluation sets used both to sweep weights and to report improvements?
5. Can you report absolute metric scores (not only normalized trends) and include a small set of direct comparisons against standard MBR and strong MBR baselines using fixed, pre-declared weights?

**Limitations:**

yes

**Strengths And Weaknesses:**

Strengths:
- The proposed `channel` view is a potentially useful lens for interpretability: it encourages thinking about which part of MBR scoring is actually driving changes for a given metric.
- The experimental section is broad in task coverage (MT, summarization, captioning) and compares multiple utilities in MT, which is helpful for teasing apart metric-dependent behaviors.
- The discussion attempts to connect and reconcile prior empirical findings about when certain MBR variants help, using the channel-weight perspective.

Weaknesses:
- The central derivation has apparent correctness issues. In particular, the step claiming equivalence from Eq. (9) to Eq. (10) introduces a square root on per-reference terms in a way that changes the objective (unless additional justification or missing terms are provided). As written, this undermines the claim that the method is a true decomposition of standard MBR rather than a new heuristic objective.
-  Turning a utility score into quantities denoted as probabilities requires explicit assumptions (non-negativity, handling of zeros, calibration, and what happens for metrics whose raw scores can be negative). This is especially relevant given the inclusion of COMET as a utility.
- Eq. (13) appears to define \(P(r_j \mid h_i)\) with a normalization that sums over hypotheses for fixed \(r_j\), which does not match the semantics of a conditional distribution over references given a hypothesis. If this is a typo, it is a serious one because it affects both the theoretical narrative and reproducibility.
- The connection to Bayes’ rule is conceptually confusing in the presence of the “swap” construction. If \(P(h_i \mid r_j)\) is defined from one directional metric score and \P(r_j \mid h_i)\) is separately defined from the swapped-direction score, these terms generally cannot both come from a single joint distribution, so the noisy-channel story needs to be reframed as a modeling choice rather than a decomposition.
- The introduction of four continuous weights raises practical concerns: selecting them without overfitting, understanding sensitivity, and reporting computational overhead. The paper does not provide runtime or cost analysis despite heavy utilities like BERTScore, and does not give guidance for choosing weights without extensive sweeps.
- Presentation needs tightening: there are multiple typos and spacing issues, and several key quantities (\(P(h_i)\), \(P(r_j)\), and the exact mapping from utility scores to “probabilities”) are not clearly defined in the main text.

---

> ### Author Rebuttal · Authors · 2026-03-31
>
> Thank you very much for your insightful comments. We appreciate your feedback and address each point below.
>
> ---
> **Weaknesses:**
> > W1: Claiming equivalence from Eq. (9) to Eq. (10)
>
> Thank you for the comment. Since P(h_i \mid r_j) is a probability distribution, its values lie in [0,1]. Therefore, applying a square root results in a monotonic transformation, yielding an equivalent objective. We will clarify this step in the final version.
>
> ---
> > W2: Treatment of negative utility values
>
> Thank you for the question. Since Eq. (6) takes an argmax, the objective is invariant under additive shifts. Thus, if utility values include negative numbers, we can shift them so that the minimum becomes zero without affecting the result. In addition, the COMET model we use (wmt22-comet-da) outputs scores in the [0,1] range and does not produce negative values, which we have verified. We will explicitly clarify this in the revision.
>
> ---
> > W3: Regarding Eq. (13)
>
> In our formulation, we define inversion as swapping f(h_i, r_j) to f(r_j, h_i). Following the same derivation as in Eqs. (7), (8), and (9), we can analogously derive P(r_j \mid h_i). We will provide a more detailed explanation of this transformation in the final version.
>
> ---
> > W4: Noisy-channel story needs to be reframed as a modeling choice rather than a decomposition
>
> Thank you. Our framework can be interpreted as a way to approximate bidirectional interactions using a fundamentally one-directional model. We will frame it more clearly as a modeling choice and revise the description accordingly.
>
> ---
> > W5: Practical concerns
>
> Our experiments perform a comprehensive sweep over channel weights to analyze the behavior of different utility metrics and tasks. We observe that while metric-specific sensitivities exist, the overall trends are stable across tasks. Based on these observations, practical guidance can be derived; for example, for BERTScore, setting \alpha and \delta relatively high and other weights lower is effective (see Findings 6 and 7).
>
> Regarding computational cost, our proposed decomposition mainly involves simple operations such as weighted sums and scaling, and can be computed efficiently in a sequential manner. As a result, it introduces negligible overhead compared to the standard MBR. We will clarify these points in the final version.
>
> ---
> > W6: Presentation needs tightening
>
> We will carefully proofread the manuscript and improve clarity. In particular, we will add explanations for key quantities, including the handling of negative values as discussed above or something.
>
> ---
> **Key Questions:**
> > Q1: Please state explicit definitions [...]
>
> As discussed in W2, since the objective involves an argmax, negative utility values can be handled via shifting without affecting the result. In practice, the metrics used in our experiments do not produce negative values. We will clarify this in the revision.
>
> ---
> > Q2: From Eq. (9) to Eq. (10)
>
> Please see our response to W1. Moreover, this formulation remains consistent with the derivation of standard MBR (Eq. (5)), ensuring equivalence of the objective. We will clarify this more explicitly in the revision.
>
> ---
> > Q3: Regarding Eq. (13)
>
> Please see our response to W3.
>
> ---
> > Q4: What is the exact selection protocol for channel weights?
>
> Our experiments are designed as a comprehensive analysis, sweeping all combinations of weights. While metric-specific differences exist, the results are almost stable across tasks (see W5 for details). In practice, the weights can be fixed to a single set of values, which we will clarify in the revision.
>
> ---
> > Q5: Can you report absolute metric scores? [...]
>
> Thank you for the suggestion. In our current presentation, we use normalized values because absolute scores vary substantially across tasks and metrics, making it difficult to visualize all configurations in a single plot while preserving readability. Since our primary goal in this analysis is to examine how performance changes across different weighting schemes, normalization provides a consistent and comparable scale.
>
> Regarding the baseline, as discussed in the Prior Advances paragraph of Section 2 (Background and Related Work), prior work has mainly focused on improving the efficiency of standard MBR. Therefore, we adopt standard MBR as a strong and widely accepted baseline. For example, when applying the best weights found on WMT22 En–De to WMT23 En–De, we observe an improvement from 50.05 (standard MBR) to 50.18, demonstrating that our reweighting yields consistent gains across settings.
>
> We will further expand such absolute comparisons in the final version to complement the normalized analysis.
>
> ---
> We hope that our responses have clarified the theoretical aspects of the derivation and addressed the experimental concerns. We will include the additional results in the final version. We believe that the main concerns have been addressed, but we would be happy to provide further clarification if needed.
>
> Best,

---

> > ### Author Rebuttal · Reviewer_6oCu · 2026-04-04
> >
> > Thank you for the response. My question has been resolved, and I have updated my scores accordingly.

---

> > > ### Author Response · Authors · 2026-04-06
> > >
> > > Dear Reviewer 6oCu,
> > >
> > > Thank you for your reply. We are very glad to hear that your questions have been resolved.
> > >
> > > We will incorporate the promised revisions into the final version. We also plan to improve the clarity of the paper by fixing typos, unifying notation, and adding additional explanations. Thank you again for your helpful feedback.
> > >
> > > ---
> > > We have one small request.
> > >
> > > In your comment, you mentioned *“I have updated my scores accordingly,”* but it appears that the scores may not have been updated from the initial rating of 3. **Could you kindly update your evaluation scores at your convenience?**
> > >
> > > Thank you very much for your time and consideration!
> > >
> > > Best,

---

### Official Review · Reviewer_CZzv · 2026-03-12

**Soundness:** 3
**Presentation:** 4
**Significance:** 3
**Originality:** 3
**Overall Recommendation:** 5
**Confidence:** 3

**Summary:**

This paper proposes a noisy-channel decomposition of MBR decoding, factoring the MBR scoring function into four probabilistic components: hyp to reference likelihood, reference to hyp likelihood, hyp prior and reference prior.
Each component is assigned a tunable weight, and the paper conducts extensive grid-search experiments across MT, summarization, and captioning to analyze which components matter most for different utility functions.

**Compliance With Llm Reviewing Policy:**

Affirmed.

**Final Justification:**

I raised my score after Author rebuttal

**Key Questions For Authors:**

In Section 3.2 you note that BERTScore and chrF are approximately symmetric under argument swapping and thus this makes P(h|r) and P(r|h) almost redundant.
Have you thought of measuring the actual correlation between these two terms across your utility matrices for these two metrics ?
If the effective rank differs for each metric that would explain the different sensitivity patterns in Figure 1.

**Limitations:**

Yes

**Strengths And Weaknesses:**

Strengths:
1) I think this paper provides a useful lens for understanding MBR decoding and an unifying framework for MAP, original MBR, conditional, inverse as special cases of the noisy channel MBR.
2) extensive empirical validation (12 MT scenarios, 3 summarization and 2 captioning).
3) well written and well organized.
4) the finding that channel weights are metric dependent but largely task agnostic is interesting.

Weaknesses:

1) The cross task claim would be stronger with a second cross task metric. In the paper only BERTScore right now is used and it shows consistent trends across tasks. I agree with authots that few metrics are genuinely comparable across tasks but maybe BLEU can be still be compared across MT and image captioning for example ?

2) I think it would add value to the paper and it would be interesting to examine/report some cases where reweighting selects a worse hypothesis than standard MBR.

---

> ### Author Rebuttal · Authors · 2026-03-31
>
> Dear Reviewer CZzv,
>
> We are sincerely grateful for the time and effort you devoted to reviewing our manuscript. We also appreciate your positive evaluation of our methodological and empirical contributions. Your valuable suggestions have helped us refine our work and address important points more effectively. We address your comments below.
>
> ---
> Weakness 1:
> > W1: The cross task claim would be stronger with a second cross task metric. [...] BLEU can be still be compared across MT and image captioning for example ?
>
> Thank you for the suggestion. Following your recommendation, we conducted additional experiments using BLEU for both captioning and translation tasks. The results (Figure A) show consistent trends across tasks, similar to those observed with BERTScore. We will include these results in the final version and add further discussion as Finding 8. Thank you again for the comments. This addition strengthens our cross-task claims.
>
> Figure A: https://i.postimg.cc/d0Df10Cg/Cross-Task-Comparison-of-BLEU-Scores-in-Machine-Translation-and-Image-Captioning.png
>
> (During the rebuttal phase, anonymized external links to figures/tables and their captions are permitted. If you are unable to access this URL, please let us know. We will provide an alternative link or consider another way.)
>
> ---
> Weakness 2:
> > W2: Cases where reweighting underperforms than standard MBR
>
> Thank you for the suggestion. For example, we observe that performance can degrade when the weight on \delta is reduced, i.e., when the reference prior is underemphasized. This behavior is already reflected in our findings. In particular, Finding 5 discusses the role of the reference prior in relation to model-based MBR methods. In the revision, we will further analyze and discuss such cases in more detail.
>
> ---
> Key Questions
> > Q1: In Section 3.2 you note that BERTScore and chrF are approximately symmetric under argument swapping and thus this makes P(h|r) and P(r|h) almost redundant. Have you thought of measuring the actual correlation between these two terms across your utility matrices for these two metrics ? If the effective rank differs for each metric that would explain the different sensitivity patterns in Figure 1.
>
> Thank you for the insightful suggestion. In our setting, BLEU and COMET are asymmetric metrics, where swapping the reference and hypothesis can change the score. In contrast, chrF and BERTScore are symmetric, as they rely on alignment-based scoring and are invariant to such swapping. Consistent with this property, we observe that symmetric metrics yield identical values for Swap and the original MBR. We will expand the comparison between these theoretical relationships (e.g., Table 1) and empirical results in the revision.
>
> Regarding the sensitivity patterns observed in Figure 1, these are likely influenced by metric-specific characteristics, such as score sensitivity and calibration behavior. This suggests that multiple factors contribute to the observed differences. Nevertheless, our results consistently show that while metric-specific behaviors exist, the overall trends are largely task-agnostic (RQ1 and RQ2). We will further clarify these observations and discuss the remaining open questions as limitations in the revision. Thank you for the helpful suggestion.
>
> ---
>
> Thank you again for your constructive feedback and valuable suggestions. We hope that our response has addressed your concerns. Your feedback has helped strengthen our claims and improve the overall clarity of the paper. We will include the additional results in the final version. If you have any further questions, we would be happy to address them.
>
> Best,

---

> > ### Author Rebuttal · Reviewer_CZzv · 2026-04-04
> >
> > I thank the Authors for the response. I am willing to raise my score

---

> > > ### Author Response · Authors · 2026-04-06
> > >
> > > Dear Reviewer CZzv,
> > >
> > > Thank you for your reply and for updating your score positively. We are very glad to hear that your questions have been resolved.
> > >
> > > We also sincerely appreciate your recognition of the strengths of our work, such as “a useful lens for understanding MBR decoding and a unifying framework,” “extensive empirical validation,” “well written and well organized,” and that the findings are interesting.
> > >
> > > We will incorporate the promised revisions into the final version. Thank you again for your thoughtful review!
> > >
> > > Sincerely,

---

### Official Review · Reviewer_h2DQ · 2026-03-13

**Soundness:** 3
**Presentation:** 3
**Significance:** 2
**Originality:** 3
**Overall Recommendation:** 4
**Confidence:** 3

**Summary:**

The paper proposes a decomposed probabilistic interpretation of MBR decoding with the goal of both improving its performance and providing better understanding of this widely used decoding strategy. The authors introduce an intuitive probabilistic decomposition of the MBR objective into interpretable components and conduct experiments that illustrate how weighting these components differently influences the behavior and performance of the resulting decoding method.

**Compliance With Llm Reviewing Policy:**

Affirmed.

**Key Questions For Authors:**

- In multiple places in the paper, the authors claim that the proposed decomposition and appropriate weighting of its components can lead to performance improvements compared to standard MBR decoding. While the ablation studies demonstrate that varying the weights associated with the different components leads to relatively consistent changes in performance, they do not directly include standard MBR decoding as a baseline in these analyses. As a result, the central claim that the proposed formulation improves over original MBR is not directly supported by the presented experiments. The authors are encouraged to conduct similar ablation studies that explicitly include standard MBR decoding as a baseline. Such comparisons would help verify whether the observed effects translate into improvements over the conventional approach and would strengthen the empirical support for the theoretical intuition developed in the paper.

- The authors demonstrate that for several commonly used asymmetric metrics, generation quality depends on directional relationships between hypotheses and references. However, the evaluation does not include comparisons with other strong alternatives such as similarity measures derived from text embedding models or metrics like ROUGE. Recent work has shown that embedding-based similarity metrics can be competitive with the metrics used in this paper when employed within MBR decoding. Including such alternatives would help assess whether the observed benefits stem specifically from the asymmetric properties of the metrics considered, or from more general properties of semantic similarity measures. In particular, embedding-based similarity functions can often be formulated in both symmetric and directional variants (e.g., through pooling or aggregation strategies), which could provide a useful test of the paper’s central hypothesis regarding the importance of directional effects. Evaluating such metrics would therefore help clarify the scope and generality of the paper’s conclusions.

**Limitations:**

Yes

**Strengths And Weaknesses:**

Their approach is elegant and well-presented. They clearly lay out their approach that decomposes the MBR decoding into well-understood and interpretable components. They then orchestrate a comprehensive ablation study that clearly showcase the individual contribution of each component, demonstrating how modeling both hypothesis-to-reference and reference-to-hypothesis interactions leads to measurable changes in the resulting performance. The experiments are thorough and help build intuition about the role of directional effects and prior terms in shaping the final hypothesis selection.

That said, some of the claims made by the authors are not fully supported by the results of the ablation study. While the theoretical framing is elegant, the empirical evidence presented does not always clearly substantiate the broader conclusions drawn, which may limit the applicability and practical relevance of the proposed approach. I elaborate on these concerns in the questions below.

---

> ### Author Rebuttal · Authors · 2026-03-31
>
> Dear Reviewer h2DQ,
>
> Thank you for your constructive feedback. We also appreciate your recognition of many strengths of our work, e.g., *“Their approach is elegant and well-presented”*. Your thoughtful and detailed comments are instrumental in strengthening our paper and clarifying our arguments. We address each question below.
>
> ---
> Key Questions:
> > Q1: Regarding including standard MBR decoding in the results
>
> Thank you for the suggestion. As shown in Table 1, the standard MBR baseline corresponds to the setting (\alpha, \beta, \gamma, \delta) = (0, 0, 1, 1), which we have verified to produce identical results. Since reweighting yields better results than the standard MBR baseline, we will expand comparisons using absolute scores (e.g., in tables) in the final version. For example, when applying the best weights found on WMT22 En–De to WMT23 En–De, performance improves from 50.05 (standard MBR) to 50.18, demonstrating consistent gains from reweighting across settings. We will also improve the plots to make comparisons with the baseline more clearly visible. These additions will complement the normalized analysis. Thank you for the helpful suggestion.
>
> > Q2: Regarding symmetric and directional variants
>
> Thank you for the insightful suggestion. In our setting, BLEU and COMET are asymmetric metrics, where swapping the reference and hypothesis can change the score. In contrast, chrF and BERTScore are symmetric, as they rely on alignment-based scoring and inherently produce identical scores under such swapping.
>
> Our ablation study already includes both symmetric (chrF, BERTScore) and asymmetric (BLEU, COMET) metrics, as well as surface-based (BLEU, chrF) and embedding-based metrics (COMET, BERTScore), providing a comprehensive evaluation across these dimensions. Furthermore, the selected metrics are widely used and considered strong evaluation metrics in machine translation. The results show that, regardless of these differences, directional weighting plays a crucial role. This highlights the importance of assigning different weights to each direction, rather than using uniform weighting as in standard MBR.
>
> We will further clarify our experimental design, the rationale behind metric selection, and the motivation for comparisons, and expand the discussion on the scope and generality of our experiments in the revision. Thank you for the helpful suggestions.
>
> ---
>
> We hope that our response has addressed your concerns. We will include the additional results in the final version. If you have any further questions, we would be happy to address them.
>
> Best,

---

> > ### Author Rebuttal · Reviewer_h2DQ · 2026-04-04
> >
> > I thank the authors for their response and consideration. That said, I do not see in the paper results supporting the claim "Since reweighting yields better results than the standard MBR baseline". The authors in the rebuttal mention that "when applying the best weights found on WMT22 En–De to WMT23 En–De, performance improves from 50.05 (standard MBR) to 50.18" which seems like a marginal improvement (in comparison to the complexity of the approach over the original MBR). I would like to see a similar thorough comparison as reported in Figures 1 and 2, but with original MBR as baseline. That would clearly illustrate the advantages of the proposed approach over original MBR and provide direct insights on how produce weightings that would consistently improve upon this powerful decoding approach.
> >
> > Concerning my second question, have the authors considered modern embedding models and similarity based their embeddings vs BERTScore?

---

> > > ### Author Response · Authors · 2026-04-06
> > >
> > > Dear Reviewer h2DQ,
> > >
> > > Thank you for your reply and feedback. We also appreciate the opportunity to further clarify our response to your comments. We address your two remaining concerns below.
> > >
> > > ---
> > > > Concern 1: I do not see in the paper results supporting the claim "Since reweighting yields better results than the standard MBR baseline".
> > >
> > > Thank you for pointing this out. We believe that the phrasing may have caused confusion. We will rephrase it to clarify this point, e.g., “reweighting can sometimes lead to better results than standard MBR.”
> > >
> > > This paper provides a unified interpretation of MBR decoding through a noisy-channel-based decomposition in the first part, and then analyzes the importance of each channel via reweighting. Therefore, the main contribution of this work is a unified interpretation based on the noisy-channel framework and trend analysis, rather than performance improvement. While we partially showed in the rebuttal that reweighting can sometimes yield higher scores than standard MBR, this was intended as a finding rather than a primary claim. Therefore, we will revise the phrasing in the paper to clarify that performance improvement is not the main objective.
> > >
> > > Furthermore, even when the improvement is marginal, the observation that simple reweighting can lead to gains is itself novel and important, as it provides insights into the principles of MBR decoding and potential directions for further development. We also believe this finding is meaningful from a decision-theoretic perspective.
> > >
> > > In addition, our experiments present global trends of trajectories in a unified manner using z-scores. Therefore, baseline comparisons do not affect the overall trajectory in the plot. If a direct comparison with the baseline is required, it would be more appropriate to present pointwise comparisons, e.g., in a table. In the final version, we will include such comparisons, e.g., using fixed best parameters and the parameters of each method shown in Table 1.
> > >
> > > To summarize, we will revise the phrasing accordingly to clarify that this is an analysis paper and that the observed improvements are one of the findings rather than the main objective. We will also present pointwise comparisons, including the baseline, in a new table.
> > >
> > > ---
> > > > Concern 2: Have the authors considered modern embedding models and similarity based on their embeddings vs BERTScore?
> > >
> > > Thank you for the comments. It is known that embedding-based methods often do not perform well, e.g., for MT [1]. While embedding models can capture coarse semantic similarities, they struggle with subtle distinctions in model-generated outputs, particularly when capturing fine-grained differences between similar texts. Therefore, such embedding models are rarely used as utility functions in MBR decoding or as evaluation metrics in text generation tasks such as MT. Instead, embedding models fine-tuned for specific downstream tasks, e.g., COMET, are commonly used as metrics or utility functions.
> > >
> > > For this reason, we employ COMET and BERTScore as embedding-based metrics in our experiments. BERTScore performs fine-grained scoring via token-level alignment, while COMET computes scores based on sentence-level embeddings of references and hypotheses. Our results show that, regardless of these differences, directional weighting plays a crucial role. In this sense, our experimental design already accounts for your concern.
> > >
> > > In the final version, we will further clarify our experimental design, the rationale behind metric selection, and the motivation for the comparisons, and expand the discussion on the scope and generality of our experiments.
> > >
> > > [1] Semantic similarity prediction is better than other semantic similarity measures (Herbold, TMLR 2024). arXiv:2309.12697. Last paragraph on p.9 (“Figure 5 shows~”). Embedding models perform well on STS tasks, but poorly on NLG tasks.
> > >
> > > ---
> > > Thank you again for your valuable feedback, which has helped strengthen the manuscript.
> > >
> > > Regarding Concern 1, we will adequately rephrase the corresponding parts and emphasize the main focus, as well as present the interpretation and merits of reweighting, e.g., using tables. Regarding Concern 2, we will clarify the comparison of the metrics used in our experiments, making it explicit that your suggestion has already been addressed in the paper through the comparison and discussion of COMET vs BERTScore. We believe these concerns can be addressed with minor revisions in the final version.
> > >
> > > We also sincerely appreciate your recognition of the strengths of our work, such as “Their approach is elegant and well-presented,” “orchestrate a comprehensive ablation study that clearly showcases the individual contribution of each component,” and “The experiments are thorough and help build intuition.” By incorporating your suggestions, we believe the paper and its claims and explanations will be further strengthened.
> > >
> > > We kindly ask you to reconsider your evaluation or provide further feedback.
> > >
> > > Best,

---

### Decision · Program_Chairs · 2026-04-30

**Decision:**

Accept (regular)

**Comment:**

This paper introduces Noisy-Channel Minimum Bayes Risk (NC-MBR) decoding, a novel and effective method for improving text generation. The key idea is to reframe MBR decoding within a noisy-channel framework, which factors MBR scoring into four interpretable components: a hypothesis-to-reference term, a reference-to-hypothesis term, a hypothesis prior, and a reference prior. Each component is assigned a tunable weight, and the paper conducts extensive grid-search experiments across MT, summarization, and captioning to analyze which components matter most for different utility functions. This framework is able to: (i) give a unifying interpretation of several existing MBR variants and (ii) analyze how channel weights interact with different utility functions and tasks.

The final reviews after the rebuttal period were mostly positive. One reviewer initially recommended weak rejection, raising various questions about the technical details. The authors provided a detailed and compelling rebuttal that successfully convinced the dissenting reviewer, who acknowledged the satisfaction and agreed to raise their score, bringing all reviewers into consensus.

Given the paper's novel empirical and conceptual contributions, and with all initial concerns thoroughly addressed, this work is a clear and good candidate for acceptance. It presents a valuable and impactful advance that will be of high interest to the community.